# Deuterium-Depleted Water in Cancer Therapy: A Systematic Review of Clinical and Experimental Trials

**DOI:** 10.3390/nu16091397

**Published:** 2024-05-06

**Authors:** Yutong Lu, Hongping Chen

**Affiliations:** 1Queen Mary School, Jiangxi Medical College, Nanchang University, Nanchang 330031, China; a18263861445@126.com; 2Department of Histology and Embryology, School of Basic Medical Sciences, Jiangxi Medical College, Nanchang University, Bayi Road 461, Nanchang 330006, China

**Keywords:** deuterium-depleted water, alternative tumour treatment, reactive oxidative stress, Keap1-Nrf2

## Abstract

Chemotherapy exhibits numerous side effects in anti-tumour therapy. The clinical experiments indicated that deuterium-depleted water (DDW) monotherapy or in combination with chemotherapy was beneficial in inhibiting cancer development. To further understand the potential mechanism of DDW in cancer therapy, we performed a systematic review. The data from experiments published over the past 15 years were included. PubMed, Cochrane and Web of Science (January 2008 to November 2023) were systemically searched. Fifteen studies qualified for review, including fourteen in vivo and in vitro trials and one interventional trial. The results showed that DDW alone or in combination with chemotherapy effectively inhibited cancer progression in most experiments. The combination treatment enhances the therapeutic effect on cancer compared with chemotherapeutic monotherapy. The inhibitory role of DDW in tumours is through regulating the reactive oxygen species (ROS)-related genes in Kelch-like ECH-associated protein 1 (Keap 1) and Nuclear erythroid 2-related factor 2 (Nrf2) signalling pathways, further controlling ROS production. An abnormal amount of ROS can inhibit the tumour progression. More extensive randomized controlled trials should be conducted to evaluate the accurate effect of DDW in Keap1-Nrf2 signalling pathways.

## 1. Introduction

In modern society with advanced medical treatment, cancer ranks second in the top ten causes of death in the United States, preceded only by heart disease [1]. As a principal source of death worldwide, cancer gave rise to nearly 10 million deaths in 2020 [2]. In terms of cancer treatments, there are currently many categories, such as surgery, radiation therapy, chemotherapy and cellular immunotherapy. Though chemotherapy can improve the survival rate of patients, numerous side effects are correlated with the therapeutic modalities for cancer, including alopecia, diarrhoea, fatigue, anorexia and anaemia [3]. With the deepening of clinical practice, adjuvant therapy has emerged. Adjuvant therapy is often used after initial treatment, such as surgery or chemotherapy, to enhance the effect of destroying cancer cells and reduce the possibility of cancer recurrence [4]. It has been identified that deuterium-depleted water (DDW) significantly strengthens the chemotherapeutic effects and attenuates side effects, which can serve as an adjuvant therapy [5].

DDW is a type of water with low deuterium atom content. Hydrogen can be divided into three different mass isotopes: protium (1H), deuterium (2H is D), and tritium (3H is T). The classification of water depends on the composition of the hydrogen isotopes. The heavy water (D20) comprises oxygen and deuterium (D) with a mass of 2 [6,7,8,9]. In natural water, the hydrogen is composed of 99.98% ordinary hydrogen (1H) and 0.015% deuterium, indicating the 150 ppm deuterium abundance, the same as standard water in people’s daily lives [9,10]. Several methods have been used to produce DDW, including desalination, distillation, catalytic exchange, etc. [11,12,13].

The usage of DDW in clinical research means a daily intake of 25 to 125 ppm concentration of deuterium atom, which replaces normal water to achieve a deuterium-depleted environment in the body. Numerous studies have reported that DDW plays a role in antitumour effects. Recent studies show that DDW inhibits the proliferation and migratory ability of numerous tumour cells, including lung, nasopharyngeal, breast and colorectal cancer [14]. Furthermore, it should be noted that DDW accelerates the apoptosis, autophagy and senescence in the tumour cells. Nevertheless, one experimental study demonstrated that DDW has no significant effect on cancer cells [15]. However, combinations of DDW with standard antitumor regimens show a striking promotion in the inhibition of proliferation, cell cycle arrest, and abnormal reactive oxygen species (ROS) production [14,16,17,18,19,20,21]. The usage of DDW in combination therapy likewise plays a vital role in decreasing the dose of chemotherapy drugs to reach the same desirable therapeutic outcomes, thus reducing the side effects of chemotherapy drugs [18].

The transcription factor, known as nuclear factor erythroid 2-related factor 2 (Nrf2), pivotal in controlling cellular defence mechanisms against oxidative stress from both external and internal sources, has been acknowledged since its identification in the 1990s. In healthy cells, Nrf2 is activated in response to oxidative stress, subsequently relocating from the cytoplasm to the nucleus. Once in the nucleus, Nrf2 initiates the transcription of various genes that contribute to the oxidative defence mechanism, thereby safeguarding against DNA damage and inflammation [22,23]. Furthermore, the vital function of the Kelch-like ECH-associated protein 1 (Keap 1)–Nrf2 signalling pathway in cancer progression is underscored by its capacity to govern the expression of a multitude of genes that influence apoptosis, cellular survival, proliferation, inflammation and the spread of tumours, thereby ensuring the survival of cancer cells [24]. Nevertheless, in terms of tumour cells, it has been acknowledged that the dysregulation of the Keap1-Nrf2 signalling pathway plays a pivotal role in numerous cancers, leading to aberrant Nrf2 activities and oxidative stress disorders, particularly in cases of cervical cancer [25]. The beneficial impact of Nrf2 on a range of neurological disorders has been progressively recognized [26], opening new avenues for therapeutic interventions by means of Keap1-Nrf2 activation.

This systematic review aims to collect and depict current clinical and experimental data on the effect of DDW on the treatment of tumours. Furthermore, these results are summarized and analysed for an overall understanding of the potential mechanism of DDW in treating cancer. This study focuses on the effects of DDW on cancer cell lines, animal cancer models and human clinical trials, demonstrating the promising application of DDW alone or combined with conventional chemotherapy for clinical trials in the future.

## 2. Materials and Methods

### 2.1. Eligibility Criteria

The inclusion criteria were as follows: (a) assessment of the effect of DDW on tumours in vivo or in vitro; (b) prospective and randomized controlled trials (RCTs); (c) studies written in English.

The exclusion criteria were as follows: (a) retroprospective clinical trials; (b) medical hypotheses; (c) non-English; (d) proceedings paper; (e) meeting abstract; (f) no original research including reviews, non-research letters, etc.

### 2.2. Search Strategy

The study was implemented under the guidelines of PRISMA. We searched three databases, including Cochrane, PubMed and Web of Science, using the keywords “deuterium depleted water”, “therapy” and “neoplasm”. All the in vivo and in vitro studies were recognized and filtered. Data were collected from 1 January 2008 to 10 November 2023. See the Appendix A, page 1, for the detailed search strategy.

### 2.3. Selection of Studies

The chosen records were input into EndNote referencing software (version 20.6, 2020). Further, two independent reviewers (L-YT and C-HP) participated in screening the titles and abstracts of those records. In the reports regarded as eligible, their full-text articles were sought for retrieval. Two reviewers jointly resolved the divergences and differences via discussion. The citations of the chosen articles were inspected to detect potentially beneficial related reports. The newest study results were selected when the same results were displayed in several studies.

### 2.4. Data Extraction

Two independent reviewers (L-YT and C-HP) used a predefined data extraction sheet to extract information. Divergence in terms of data extraction was resolved via discussion. The studies’ details included the following: year of publication, nation, study design, participants, the total size of the sample, category of the tumour, intervention details and main findings.

## 3. Results

### 3.1. Study Search and Selection Result

In the search process, 648 studies were selected from three databases. Among them, 103 were removed because of duplication. In screening, 519 were excluded by their relevant titles and abstracts. Furthermore, 26 studies were evaluated for eligibility by reading the full contents, 12 of which were excluded for the following reasons: (a) non-randomized controlled trials (non-RCTs) (*n* = 2); (b) medical hypotheses (*n* = 1); (c) non-English (*n* = 4); (d) proceedings paper (*n* = 1); (e) meeting abstract (*n* = 3); (f) review (*n* = 1). Our systematic review included 15 studies. In addition, one study was identified by searching the references of the relevantly selected studies. In summary, 15 studies were included in this systematic review, among which 14 were in vitro and in vivo studies, and 1 was an intervention study. A study flowchart is depicted in Figure 1 [27].

### 3.2. Study Characteristics

Of the included studies, two were conducted in China [28,29], five in Hungary [19,30,31,32,33], three in Iran [15,17,21], two in Romania [16,34], one in Sweden [18], one in Turkey [20], and one in Russia [35]. The studies identified were categorized into two types based on their subject: fourteen studies of the effect of DDW on cancer cell lines and animal cancer models [15,16,17,18,19,20,21,28,29,30,31,32,34,35] and one clinical study in humans [35].

This article summarizes the existing research results on treating nine common cancers with DDW, focusing on the experimental design and results reported in the literature. Due to the limited number of relevant available studies, we did not conduct a meta-analysis of specific outcome measures. However, existing research has demonstrated the promising impact of DDW on cancer treatment.

### 3.3. Experimental Trials Using DDW on Cancer Cell Lines and Animal Cancer Models

In the relative research, almost all experimental trials exert positive effects of DDW on tumour cells and animal cancer models. Furthermore, combining DDW and chemotherapy demonstrates a promising impact on the prognosis. Hence, the usage of DDW in treatment could be separated into two parts: single usage of DDW and combination of DDW and chemotherapy drugs. The experimental results for DDW in this review are presented in Table 1 and Table 2 and then analysed.

#### 3.3.1. DDW Monotherapy in Cancer

##### The Effect of DDW on the Migration and Proliferation of Tumour Cells

The association between the migration and proliferation of tumour cells and DDW was exhibited in nine experiments. In eight of the included studies, it has been observed that DDW can inhibit the proliferation of tumour cell lines and animal tumour models in mice compared with normal water [17,18,19,21,28,29,32,35]. However, one experiment exhibited no cytotoxic effect on tumours in the DDW medium [15]. Another study showed that DDW can suppress the migratory capability of cancer cells [32].

**Table 1 nutrients-16-01397-t001:** Summary of studies investigating the effects of DDW on cancer cell lines and animal cancer model.

Nation	Type of Cells/ Type of Animals	Type of Cancer	Main Results	Reference
China	A549 HLF-1	Human lung carcinoma	Suppresses the growth of A549 cell lines; observes myelin bodies and physalides in the cytoplasm; increases S phase, reduces G0 to G1 and G2 to M phases. Induces cell apoptosis.	[28]
	H460 Male BALB/c nude mice (two groups: control, DDW)	Human lung carcinoma	Observes an obvious decrease of 30.80% in tumour inhibition rates.	
Hungary	Male CBA/Ca mice (four groups: 150 ppm, 150 ppm + DMBA, 25 ppm, 25 ppm + DMBA)	Mice lung carcinoma	No significant elevation of the of the expression of the Bcl2, Kras and Myc genes by DMBA.	[30]
	Female CBA/Ca mice (four groups: 150 ppm, 150 ppm + DMBA, 25 ppm, 25 ppm + DMBA)	Mice lung carcinoma	Decreases the upregulation of the Bcl2, Kras and Myc gene expression by DMBA.	
China	CNE-1 CNE-2 5-8F 6-10B Sune-1 preosteoblast MC3T3-E1	Human nasopharyngeal carcinoma	Inhibits the proliferation of NPC cell lines; colony formation was promoted in normal preosteoblast MC3T3-E1 cells and was markedly inhibited in NPC tumour cells; increases G1 phase, reduces S phase in NPC cell lines; markedly inhibits migration in CNE-2, Sune-1 and CNE-1 cells; promotes NQO1 protein expression in NPC cell lines; significantly decreases the expression of PCNA and MMP9 in NPC cells.	[29]
Iran	MDA-MB-231 PC-3 HCT-116 U-87MG AGS HDF-1	Human breast adenocarcinoma Human prostate adenocarcinoma Human colon carcinoma Human glioblastoma multiforme Human gastric adenocarcinoma	No cytotoxic effects on all cell lines in DDW monotherapy.	[15]
Romania	MDA-MB-231	Human breast adenocarcinoma	No significant difference in mitochondrial membrane potential and nuclei integrity; slightly promotes autophagy and senescence; upregulates 528 and downregulates 368 miRNAs; upregulates MiR-155, MiR-205, downregulates MiR-210, MiR-181a/b/c, MiR-200 family members (miR-200b, miR-429 and miR-141) and let-7b family members.	[16]
Romania	DLD-1	Human colorectal carcinoma	Activates senescence, upregulates 46 and downregulates 59 miRNAs; upregulates let-7b and downregulates miR-23b and miR-193b.	[34]
Iran	MCF-7	Human breast adenocarcinoma	Inhibits cell growth; 50ppm DDW alone stops tumour proliferation at the G0/G1 stage and decreases the S phase.	[17]
Sweden	MCF7 A549 HT29	Human breast adenocarcinoma Human lung carcinoma Human colorectal adenocarcinoma	Inhibits A549 cells most efficiently; downregulates p53 signalling; suppresses glutathione metabolism pathways; DDW arrests cells in late S and G2 phases; increases ROS amount in the cells grown in 80ppm DDW for 48 h; promotes proteome thermal stability.	[18]
Hungary	MIA-PaCa-2 H449 MCF-7	Human pancreatic carcinoma Human lung carcinoma Human breast carcinoma	Decreases synthesis and turnover of new RNA ribose in MIA-Paca-2; decreases synthesis of nuclear membrane cholesterol in MIA-Paca-2; exhibits a dose-dependent inhibition of growth rate of MIA-Paca-2; significantly decreases the production of G6PDH flux and NADPH.	[19]
Turkey	EAT, male BALB/c mice (four groups: 150 ppm, 150 ppm + EAT, 85 ppm, 85 ppm + EAT)	Mouse mammary adenocarcinoma	Upregulates GSH, CAT, Na+/K+-ATPase and PON1 in tumour + DDW group compared to tumour group; downregulates LPO, GPX, GR, GST, GGT, PC, SDH, ALT, AST, MPO and XO in the tumour + DDW group compared to the tumour group.	[20]
Hungary	A459	Human lung carcinoma	Higher Deuterium/Hydrogen Ratio stimulates cancer-related and kinase genes expression, DDW upregulates 1 cancer-related gene, downregulates 5 cancer-related genes and 1 kinase gene.	
	Male CBA/Ca mice (two groups: 150 ppm + DMBA, 25 ppm + DMBA)	Mice lung carcinoma	Significantly improves 1-year survival.	[31]
	Female CBA/Ca mice (two groups: 150 ppm + DMBA, 25 ppm + DMBA)	Mice lung carcinoma	Significantly improves 1-year survival.	
Hungary	4T1, female and male BALB/cJ mice (six groups: CTRL water, CTRL yolk, DDW, DDW + DDOC, DDOC, DU283)	Mice mammary carcinoma	DDW and DDyolk decrease primary tumour size and weight of metastasis.	[32]
	MCF-7, female and male NSG immunodeficient mice (three groups: control, DDyolk-treated, DDyolk- and DDW-treated)	Human breast carcinoma	DDW and DDyolk increase the survival time of mice; slightly decrease tumour size.	
Iran	HT-29	Human colorectal adenocarcinoma	Inhibits the proliferation of HT-29 cells.	[21]
Russia	Murine melanoma B16, male C57Bl/6 mice (three groups: 50 ppm [D]/0, 50 ppm [D]/-30, 146 ppm [D])	Mice melanoma	The mice consuming DDW 30 days prior to inoculation showed a significant increase in survival, stronger inhibition of cancer growth and metastasis; the mice receiving DDW since tumour inoculation had no difference from control group in survival and metastasis of cancer cells and demonstrated a peak of tumour inhibition from the 20th to the 25th day following a subsequent decrease.	[35]

According to the included studies, three studies suggested that a dose-dependent inhibition of the growth of tumour cells occurred in the DDW treatment group [19,21,29]. It has recently been discovered that DDW promoted the inhibition of the colony formation and proliferation capability of nasopharyngeal tumours. Interestingly, DDW promoted the growth of normal preosteoblast cells instead of suppressing them, thereby showing a specific inhibition of tumour cells [29]. Additionally, it is worth mentioning that the 80ppm DDW had maximum suppression of tumour growth compared with both lower and higher D-concentration [18].

The effect of DDW is affected by the time of tumour inoculation. In vivo experiments indicated that the mice consuming DDW 30 days before tumour inoculation showed significant inhibition of cancer growth. In contrast, the mice receiving DDW since tumour inoculation had no difference from control in the cell growth and showed a peak in tumour inhibition until the 20th to the 25th day, following a decrease [35]. It has been indicated that no cytotoxic effect on the proliferation of six tumour cell lines was observed in the DDW medium. In contrast, the mice receiving DDW since tumour inoculation showed no effect [15].

In four studies of tumour cell lines, it has been indicated that the cell cycle changed in the DDW-treated cells [17,18,28,29]. Wang and Yavari found that the S phase in the cell cycle is decreased, and an increasing number of cells are arrested at G0 or G1 phases [17,29]. The other two experiments demonstrated that with the treatment of DDW, most cells are arrested at the S and G2 phases [18,28]. Relative research indicated that the deuterium concentration inside the cells might be a significant factor in cell division. A low deuterium concentration will arrest the growth of cells until the cells take a long time to obtain the appropriate ratio of D/H [17,18,21]. These arrested cell cycles indicate that tumour growth is inhibited in the low deuterium concentration condition.

As an important feature of cancer, cancer metastasis is the major cause of death [36]. Experimental studies have demonstrated that DDW significantly suppressed the migration of three tumour cell lines [29]. It has been shown that using DDW and deuterium-depleted yolk (DDyolk) in the daily feeding of mice decreased the weight of cancer metastasis [32]. Furthermore, it was reported that cancer metastasis was significantly inhibited in the mice taking DDW 30 days earlier [35].

##### The Effect of DDW on Apoptosis, Autophagy and Senescence in the Tumour Cells

Tumours always originate from the diminished apoptosis of cells, causing immortal malignant cells. Inducing apoptosis, autophagy or senescence in the tumour cells has become a prospective target of malignant tumour treatment [37]. Three experimental studies hint that DDW has a positive effect on promoting autophagy, apoptosis or senescence [16,28,34]. In contrast, one research study indicates no significant changes in apoptosis [15].

A beneficial elevation in apoptosis was found in the DDW group. Adding DDW significantly increases the fragmented DNA in the tumour cells [28]. Nucleus fragmentation is a remarkable hallmark of apoptosis [38]. It has been exhibited that using DDW alone could induce slight autophagy and senescence in the breast tumour, while no significant difference has been found in the integrity of nuclei in the DDW group compared with normal water [16]. In addition, it was shown that the β-galactosidase expression in the DDW medium was increased compared with the standard water medium, demonstrating the senescence of colorectal cancer cells in the DDW groups [34].

##### The Effect of DDW on the Expression Level of Cancer-Related Genes in Tumour Cells

In this systematic review, 13 genes were analysed in detail, including 10 microRNA (miRNA), as discovered in the experiments. Among them, eleven genes, including eight miRNAs, were downregulated, and three miRNAs were upregulated. A total of four downregulated genes (Myc, miR-210, 181a/b/c, 205 and 193b) and two upregulated genes (miR-205 and let-7) were correlated with a suppression in proliferation and migration of cancer. It was also indicated that one upregulated gene, miR-155, and one downregulated gene, let-7, related to increased proliferation and migration in tumours. It was found that one downregulated gene, miR-193b, and one upregulated gene, miR-155, participated in suppressing apoptosis, autophagy and senescence. It is shown that the downregulation of three genes (miR-210, 23b and Bcl-2) was linked to promoting apoptosis, autophagy and senescence in cancer [16,30,34]. Overall, eight genes were linked to affecting the proliferation and migration of tumours, and five genes were related to regulating autophagy, autophagy and senescence in cancer.

As a small, non-coding RNA, microRNA plays a critical role in silencing particular genes via microRNA-induced silencing complex (miRISC) or mediating translational activation [39]. Most malignant tumours are always connected with deviant miRNA expression levels in the cells, inducing uncontrolled proliferation and suppressing the apoptosis trend. Thereby, miRNAs are always regarded as a vital marker of cancer occurrence [40]. An upregulation of 528 miRNAs and downregulation of 368 miRNAs was found in the DDW treatment compared with the SC condition. Among them, miRNA microarray analysis showed that MiR-155 and MiR-205 are upregulated, and MiR-210, MiR-181a/b/c, MiR-200 family members (miR-200b, miR-429 and miR-141) and let-7b are downregulated in the DDW medium [16]. In another study, upregulation of 46 and downregulation of 59 miRNA was found in the DDW treatment. Let-7b was upregulated, while MiR-23b and MiR-193b were found to be downregulated in the DDW medium [34]. It has been shown that DDW upregulates 1 cancer-related gene and downregulates 5 cancer-related genes and 1 kinase gene among the 236 cancer-related and 536 kinase genes. In contrast, deuterium-enriched water (DEW) upregulates 97.3% of genes characterized by both changes in expression over 30% and numbers of copies more than 30. This indicates that the effect of DDW on the expression level of cancer-related and kinase genes is insignificant compared with deuterium-enriched water [31]. In vivo experiments indicated that the expression level of Bcl2, Kras and Myc genes was downregulated in female mice treated with DDW [30]. Most of these changes in miRNA expression caused by DDW showed that DDW could inhibit cell proliferation, invasion and metastasis and increase the sensitivity to therapeutic drugs.

##### DDW Regulates the Expression of Proteins in the Tumour

In this systematic review, 21 proteins were analysed in detail. Among them, 16 proteins were observed to be downregulated, and 6 proteins were upregulated. One downregulated protein, matrix metalloproteinase-9 (MMP9), and two upregulated proteins (NADPH: quinone oxidoreductase-1 (NQO1) and Sodium/potassium ATPase (Na+/K+-ATPase)) were correlated with a suppression in the proliferation and migration of cancer. It was also indicated that one downregulated protein, P53, was related to increased proliferation and migration in tumours. It was found that one downregulated protein, proliferating cell nuclear antigen (PCNA), and one upregulated protein, NQO1, participated in promoting apoptosis [29]. Additionally, it was shown that the downregulation of five proteins (glucose-6-phosphate dehydrogenase (G6PDH), nicotinamide adenine dinucleotide phosphate (NADPH), glutathione peroxidase (GPX), glutathione reductase (GR) and glutathione S-transferase (GST)) was linked to promoting ROS production. It was indicated that four upregulated proteins (paraoxonase 1 (PON1), reduced glutathione (GSH), catalase (CAT) and NQO1) and two downregulated proteins (myeloperoxidase (MPO) and xanthine oxidase (XO)) correlated with diminished ROS production. Furthermore, two downregulated proteins (lipid peroxidation (LPO) and sorbitol dehydrogenase (SDH)) decreased the risk of tumour occurrence [20]. Overall, four proteins were linked to affecting the proliferation and migration of tumours, and three proteins were related to regulating autophagy in cancer. Eleven proteins participated in the production of ROS and two proteins were correlated with tumour risk.

It has been shown that NQO1 expression is enhanced, while MMP9 and PCNA expression is downregulated with the declining deuterium concentration in nasopharyngeal carcinoma cells (NPCs) [29]. Another experiment showed that the abundance of P53 is decreased in the DDW medium. P53 is a tumour suppressor, playing a role in DNA repair and induction of apoptosis and cell cycle arrest [18]. Downregulation of P53 expression in the DDW may decrease the effect on the inhibition of tumour progression.

Overall, the changes in protein expression levels hint that DDW may inhibit cancer cell apoptosis and block the cell cycle.

#### 3.3.2. Chemotherapeutic Drugs in Combination with DDW in Tumours

Chemotherapy is considered one of the most critical ways to treat tumours. With the development of chemoresistance, the response rate of cancer to chemotherapy drugs decreases and the probability of recurrence increases [41]. Therefore, new drugs to treat cancer need to be explored. Over the past few years, the combination of DDW with chemotherapeutic drugs has shown promising anticancer effects via regulating specific signalling pathways [15,17,18,19,34]. In the pooled studies, six kinds of chemotherapeutic drugs were used in combination with DDW in eight studies, including paclitaxel, cisplatin, 5-Fluorouracil (5-FU), oxaliplatin, auranofin and crocin. Among them, 5-FU and cisplatin were used separately twice in the studies. The characteristics of the combination treatment studies are presented in Table 2.

**Table 2 nutrients-16-01397-t002:** Summary of studies investigating the effects of DDW and chemotherapy combination on cancer cell lines and animal cancer models.

Nation	Type of Cells/ Type of Animals	Type of Cancer	Medicine	Main Results	Reference
Iran	MDA-MB-231 PC-3 HCT-116 U-87MG AGS HDF-1	Human breast adenocarcinoma Human prostate adenocarcinoma Human colon carcinoma Human glioblastoma multiforme Human gastric adenocarcinoma	Paclitaxel	Paclitaxel remarkably decreased the surviving proportions of all cell lines. DDW enhanced the inhibition of paclitaxel on AGS, PC-3 and U-87MG but did not significantly affect HCT-116 and HDF-1.	[15]
Romania	MDA-MB-231	Human breast adenocarcinoma	Cisplatin	DDW has a weak synergistic effect on mitochondrial activity and autophagy activation in the case of cisplatin treatment of MDA-MB-231 cells.	[16]
Romania	DLD-1	Human colorectal carcinoma	5-FU Oxaliplatin	DDW shows a weakly synergic effect on pro-apoptosis in 5-FU- and oxaliplatin-treated DLD-1 cells.	[34]
Iran	MCF-7	Human breast adenocarcinoma	5-FU	5-FU inhibits MCF-7 in a concentration-dependent way; DDW enhances inhibition of 5-FU in a concentration-dependent manner; DDW enhances the decreasing proportion of MCF-7 cells at the S and G2 to M phases and the increasing ratio of MCF-7 cells at the G0 to G1 phase in the cells treated with the 5-FU combinations; DDW reverses the decrease in SOD and CAT and increase in MDA in MCF-7 cells treated with 5-FU.	[17]
Sweden	MCF7 A549 HT29	Human breast adenocarcinoma Human lung carcinoma Human colorectal adenocarcinoma	Auranofin	Combination of auranofin and 80 ppm DDW increases the ROS amount significantly compared with monotherapy of either auranofin or DDW; DDW lowers the concentration of auranofin to achieve the same suppression of cell growth as higher concentration of auranofin.	[18]
Hungary	MIA-PaCa-2 H449 MCF-7	Human pancreatic carcinoma Human lung carcinoma Human breast carcinoma	Cisplatin	DDW enhances the cell growth inhibitory effect of cisplatin on MIA-PaCa-2 in a concentration-dependent manner.	[19]
Iran	HT-29	Human colorectal adenocarcinoma	Crocin	Crocin inhibits the growth of HT-29 in a concentration-dependent way; DDW and crocin had concerted results on the inhibition of tumours; 1 mg/mL crocin in combination with 75 ppm DDW had the most significant inhibition on the proliferation of HT-29 cells at 48 h, and the combination therapy enhances the decrease in SOD and catalase and the increase in MDA compared with 75 ppm DDW alone; crocin and DDW have a synergistic effect in increasing the cell amount at the G0 to G1 phases and decreasing the cell number at the S and G2 to M phases on HT-29 cells.	[21]

##### Mitotic Inhibitors—Paclitaxel

The mechanism of paclitaxel is to induce abnormal cell division by promoting the stable β-tubulin heterodimer assembly continuously, further suppressing the process of depolymerization. As a result, the cell cycle is arrested in the G2/M phase, causing apoptosis [42].

It has been shown that DDW enhances the inhibition effect of paclitaxel on U-87MG, PC-3 and AGS tumour cell lines, which indicates that paclitaxel in combination with DDW has a potential therapeutic benefit in increasing the suppression of proliferation of tumour cells compared with paclitaxel monotherapy [15].

##### Antimetabolites—5-FU

The most crucial action of 5-FU is serving as a thymidylate synthase inhibitor, playing a role in suppressing the synthesis of thymine, thereby blocking the cell cycle and causing cell apoptosis [43].

It has been indicated that DDW, in combination with 5-FU, shows a weakly synergic effect on pro-apoptosis of tumour cells [34]. Another experiment shows that DDW enhances the inhibition of 5-FU on tumours in a concentration-dependent pattern. Additionally, it arrests MCF-7 cells at the G0 to G1 phase. The decrease in SOD and CAT and the increase in malondialdehyde (MDA) are reversed by DDW in MCF-7 cells treated with 5-FU, indicating a decreased generation of ROS. Combination therapy showed a significant effect on anti-cancers than either agent alone [17].

##### Alkylating Agents

As the earliest chemotherapy drugs, alkylating agents demonstrate effective outcomes in the treatment of tumours, particularly in lymphomas and leukaemia [44]. In this systematic review, two categories of alkylating agents, oxaliplatin and cisplatin, are exhibited.

The mechanism of oxaliplatin is inducing DNA intra-strand, inter-strand and DNA–protein crosslinks, thereby damaging the DNA to cause cell apoptosis [45]. It has been demonstrated that DDW weakly enhanced the pro-apoptosis effects of oxaliplatin on DLD-1 cells [34].

Cisplatin crosslinks DNA and interferes with mitosis to cause arrest of cell division, resulting in tumour cell apoptosis [46]. It has been shown that DDW weakly synergizes the effect of cisplatin on mitochondrial activity and activation of autophagy [16]. Additionally, DDW enhances the inhibitory effect of cisplatin on MIA-PaCa-2 in a concentration-dependent manner [19]. The combination therapies promoted autophagy and inhibited the proliferation of cancers.

##### Gold Salt—Auranofin

As a pro-oxidant agent, auranofin interferes with the intracellular redox system by inhibiting thioredoxin reductase (TrxR), increasing the production of ROS, thereby stimulating apoptosis [47].

It has been indicated that the ROS level significantly increased with the combination of auranofin and 80 ppm DDW compared with the exclusive usage of auranofin. Furthermore, the DDW intake lowers the concentration of auranofin to achieve the same suppression of cell growth as a higher concentration of auranofin, which is beneficial for decreasing the side effects of auranofin [18].

##### Plant Extracts—Crocin

It is reported that crocin can promote the apoptosis of tumour cells by upregulating Bax and P53, activating caspase-8, and downregulating the expression level of Bcl-2 [48].

It has been shown that DDW coordinates crocin on the inhibition of tumour growth and arrest of cells at the G0 and G1 phases in the cell cycle. Of note, a combination of 75 ppm DDW and 1 mg/mL crocin enhances the increase in MDA and decrease in SOD and catalase compared with 75 ppm DDW when given solely, which indicates the increasing production of ROS [21]. This result implies that combination therapy regulates cell cycle arrest, inhibits proliferation and increases ROS production synergistically.

### 3.4. Clinical Study Using DDW in the Treatment of Cancer

To date, there have been few clinical studies on the use of DDW to treat cancer patients, but with only one randomized clinical trial. However, it exhibited a significant clinical value in the prognosis of tumours, indicating that more randomized clinical trials of DDW on cancer should be conducted in the future.

In the randomized Phase II clinical trial, 44 pancreatic cancer patients who received the same conventional treatment and two different waters were involved. In the treated group, 22 pancreatic cancer patients were allocated 85 ppm DDW, while the other 22, in the placebo group, were allocated 150 ppm normal water. Furthermore, the clinical trial results showed seven patients gaining a partial response in the treated group, compared with one placebo group patient. The development of the net prostate volume decrease in the treated group was three times higher than the placebo group. Furthermore, both the number of patients with cessation of urination complaints and the one-year survival rate in the treated group were higher than in the placebo group. The decrease in net prostate-specific antigen (PSA) value was 326.1 ng/mL in the treated group compared with 243.6 ng/mL in the placebo group, with an initial PSA value of 406.4 ng/mL in the treated group and 521 ng/mL in the placebo group [33].

### 3.5. Effects of DDW on the Cellular Redox Balance in the Tumour

Oxidative stress is when the amount of ROS and antioxidant systems is imbalanced. An abnormal number of ROS in the body microenvironment will cause cellular redox disbalance, associated with the occurrence of tumours. It has been shown that ROS is necessary for inducing cancer development [49].

The measurement method of ROS in the tumour can be divided into two parts: the expression of ROS-related proteins and the direct production amount of ROS. To explore the protein expression under DDW monotherapy, 11 proteins, including G6PDH, NADPH, GPX, GR, GST, NQO1, GSH, CAT, PON1, MPO and XO, have been proved to be correlated with the production of ROS. One study indicated that GSH, CAT and PON1 are upregulated, while GPX, GR, GST, MPO and XO are downregulated in the DDW treatment. The changes in those regulatory factors show an overall decreased number of ROS [20]. Additionally, the combination of auranofin and DDW therapy exhibited elevated ROS amounts, indicating an enhancement in the formation of ROS compared with monotherapy [18]. In one DDW and 5-FU combination experiment, the increase in the SOD and CAT and the decrease in the MDA indicate the downregulation of ROS [17]. In summary, one research study hints that DDW increases ROS production, which promotes the inhibition of proliferation [18]. Of note, two research studies indicate that the amount of ROS decreases in the DDW medium [17,20].

As depicted in Figure 2, the production of ROS is regulated by several signalling pathways. Nrf2 is a vital transcription factor that is essential in promoting the expression of antioxidative genes, including GST, CAT, SOD, GSH, GR, GPX, NQO1, NADPH, G6PDH and PON1 [20,29]. As a repressor protein in the cytoplasm, Keap1 binds to Nrf2 in the cytoplasm, preventing it from entering the nucleus to control gene expression. Once the oxidative stress (ROS) increases, it will inhibit Keap1 and release Nrf2 into the nucleus. Nrf2 triggers the expression of antioxidative genes, further inhibiting ROS formation [50]. In the experiments, it has been indicated that DDW affects the expression of antioxidative genes and controls the formation of ROS. Additionally, it has been shown that DDW inhibited the oxidative genes MPO and XO, preventing ROS production [20]. In total, DDW plays a role in regulating the antioxidative and oxidative systems that affect the production of ROS.

As exhibited in Figure 3, the amount of ROS is tightly associated with tumour formation. In the specific threshold, the elevation of ROS induces the activation of tumour-related signalling pathways, thus promoting tumour formation. Nonetheless, due to the increased ROS amount in tumour cells, they are more sensitive than normal cells to further elevation of ROS. Thus, the increase in ROS exceeding the threshold could inhibit the tumour growth, triggering programmed cell death [51]. It has been suggested that DDW plays a role in changing the amount of ROS formation in the tumour and inhibits it.

In summary, DDW either promotes or inhibits ROS production in the tumour cells, significantly inhibiting tumour progression.

## 4. Discussion

This study chooses DDW to conduct a systematic review in cancer treatment because, in addition to its theoretical rationality, many aspects of DDW research in this area need to be further elaborated. Although one research study has shown the correlation of the anticancer effects of DDW on tumours with oxidative stress in the past [18], the specific molecular mechanism and signalling pathways are unclear. We pioneered the discovery that the Keap1-Nrf2 signalling pathway is an important pathway for controlling oxidative stress in DDW therapy, in which multiple oxidative stress-related genes play a role in controlling ROS in this signalling pathway. DDW controls the level of oxidative stress by controlling the expression of oxidative genes and antioxidant genes in the Keap1-Nrf2 signalling pathway, further inhibiting tumours. The discovery of the Keap1-Nrf2 signalling pathway will help to study the effect of DDW on cancer more accurately and facilitate further research on the tumour suppressor mechanism of DDW in the future. Moreover, DDW is convenient when used in tumour suppression, which is hoped to become a new means of suppressing tumours and benefit more tumour patients.

The anticancer effects of DDW have a long history. A study as early as 1993 indicates that proliferation of L929 fibroblasts and tumours in xenograft mice was significantly inhibited after treatment with DDW [52]. Regarding the promising application potential of DDW in treating tumours, numerous animal experiments and clinical trials have investigated the effect of DDW on cancer therapy in the past 30 years [15,16,17,18,19,20,21,28,29,30,31,32,34,35]. identified The beneficial clinical significance of DDW has been identified, suggesting that DDW can function as an adjuvant therapy in treating cancer. Notably, it has been indicated that combined with chemotherapy, it enhances the effects of tumour cells treated with DDW. In this systematic review, the application of DDW in tumour therapy is divided into DDW monotherapy and DDW in combination with chemotherapy to research the difference between them in the treatment outcomes.

In the DDW therapy, when used solely, most experiment studies have confirmed the evident role of DDW in cancer treatment, including significant inhibition of proliferation, metastasis and cell cycle arrest in the tumour cells. Regarding the inhibition pattern of tumours, it is worth noting that a study exhibited a maximum inhibition in the middle DDW concentration instead of dose-dependent inhibition. It has been found that when the concentration of deuterium in the medium was adjusted to be lower than that in the mitochondria matrix, it triggered an increase in the potential of the mitochondrial membrane, inducing the excessive generation of ROS and inhibiting the growth of cells. Further reduction in deuterium concentration in the medium will cause an activation of the feedback loop, resuming the ROS generation and cell growth situation to a normal level [18].

The gene expression profile can indicate the effects of DDW on the tumour treatments. It was reported that many genes were involved in the tumorigenesis. Bcl-2 is a vital gene that suppresses pro-apoptotic signals to prevent cell apoptosis, and its excessive expression will cause out of control cell proliferation [53]. As the most easily mutated proto-oncogene, the Kras gene mutation always participates in adenocarcinoma progression [54]. The overexpression of the Myc oncogene also significantly induces excess cell proliferation [55]. Suppressing these genes with DDW treatment can promote the apoptosis of tumour cells, inhibit proliferation and prevent the progression of the tumour [30]. Let-7b is beneficial in inhibiting tumour growth and progression, and low expression of Let-7b induces a worse prognosis [56]. The expression levels in cancer cells regarding Let-7b are contradicting in two experiments. One study showed downregulation of Let-7b expression, while Let-7b was upregulated in another study [16,34]. Low expression of Let-7b indicated decreased inhibition of tumour growth by DDW.

The ROS induced by DDW has become a focus in the anticancer research. Medium concentrations of ROS can induce DNA damage and instability of the genome, leading to the formation of numerous oncogenic mutations, thereby promoting tumour progression. However, high concentrations of ROS can increase oxidative injuries and ROS-dependent death signals, causing cancer cell apoptosis [51]. In the studies in this systematic review, both reduced and increased ROS production caused by DDW suppressed the cancer, indicating the promising anticancer effects.

Specific protein expression levels reflect the generation levels of ROS. It has been shown that NQO1, GSH, PON1, GPX, GR, GST and NADPH play an essential role in the central ROS regulatory system to regulate the growth of malignant cells [20,29]. NQO1 can reduce quinone to hydroquinone through a catalytic reaction. When NQO1 decreases, excess quinone forms semihydroquinone through a reduction reaction, which then generates ROS in the redox cycle [57]. GSH is an antioxidant peptide, which is significant in clearing ROS to avoid oxidative injuries [58]. PON1, as an HDL-binding protein, participates in eliminating oxidants [59]. Additionally, GPX and GR also function as antioxidants to decrease cell oxidative stress [60,61]. GST inhibits the Jun N-terminal kinase, thus protecting cells against hydrogen peroxide-induced cell death [62]. As an anabolic-reducing agent, NADPH plays a critical role in the synthesis process of reduced GSH, thereby eliminating the ROS to sustain the viability of cells in the counter of oxidative stress [63]. G6PDH is frequently upregulated in numerous tumours; lack of it increases the risk of suffering harm from oxidative stress [64]. Downregulation of both NADPH and G6PDH suppresses the protection of tumour cells from oxidative injury, which is beneficial for inhibiting the proliferation of malignant cells. The Keap1-Nrf2 signalling pathway plays a role in regulating oxidative stress in tumours. When ROS is in excess, the Nrf2 will be separated from Keap1, entering the nucleus and promoting the expression of antioxidative genes [50]. It has been indicated that DDW regulates oxidative stress by inhibiting oxidative genes and controlling the antioxidative gene expression [17,18,20]. DDW might affect Keap1 or Nrf2 expression, further regulating the antioxidative systems.

One challenge in treating tumours is resistance to chemotherapy, which can cause recurrence and more severe progression of diseases [65]. It is indicated that inducing autophagy can increase the sensitivity of tumour cells to chemotherapy, preventing tumour recurrence after chemotherapeutic treatment [66]. In previous studies, it was identified that DDW participated in the induction of apoptosis and autophagy of tumours, which is promising in reducing resistance to chemotherapy through combination therapy. Thus, DDW has the potential to reduce chemotherapy resistance, achieving a better therapeutic outcome.

In the seven experiments, the combination of DDW and six types of chemotherapy drugs significantly enhanced anti-tumour activity compared to chemotherapy drugs or DDW alone. Despite one experiment demonstrating no effect on the inhibition of cancer growth with a single usage of DDW, combination therapy with paclitaxel showed that DDW increased the inhibitory properties of chemotherapy [15]. Thus, the inhibition effect against cancers showed a synergetic promotion in combining DDW with chemotherapy. It is well known that chemotherapy has numerous side effects, causing damage to the body and disrupting the daily routine of patients [67]. The usage of DDW can decrease the dose of drugs to achieve the intended effects, further decreasing the side effects induced by chemotherapy.

Despite the overall significant outcomes of DDW in the cell lines and animal cancer models, human clinical trials are needed to verify the effectiveness, safety and potential side effects of DDW on cancer patients. This systematic review includes one double-blinded randomized controlled trial (RCT) because most DDW clinical trials are retroprospective and not RCTs. The RCT showed that more patients receiving DDW in their treatments exhibited partial response than the placebo group. Additionally, the decrease in the tumour marker and the increase in the survival condition in the DDW treatment group indicated a promising therapeutic effect of DDW.

## 5. Advantages and Limitations

The study comprehensively reviewed all experimental and clinical trials in the internationally published literature from 2008 to mid-November 2023. Based on their properties, these studies were separated into those related to the effects of DDW monotherapy on tumour cell lines and animal cancer models and the consequences of combined use with chemotherapy on tumours. The research on DDW in clinical human experiments was also mentioned separately. The safety and efficacy of DDW have been previously evaluated in different types of cancer cells and patients, and many studies have been conducted on the role of ROS in DDW treatment. On the top of that, this is the first systematic review explicitly focusing on the use of DDW in patients with cancer. For the first time, we systematically divided the use of DDW in cancer treatment into the use of DDW alone and the use of DDW in combination with chemotherapy. We split the experimental subjects into cell lines, animal cancer models and human experiments for different experimental stages. In addition, we discovered the anti-cancer effect of DDW and the production of ROS through gene and protein expression. In the related research on ROS, we found a correlation between ROS production and the therapeutic effect of DDW on cancer. However, the current systematic review has several limitations when interpreting the results, including the following: (i) the relatively limited amount of included studies, so the interpretation of results should be viewed with caution; (ii) only studies published in English were included; (iii) there are too few clinical trials of DDW in cancer patients, which may cause bias; (iv) it cannot be excluded that the dose of DDW used in clinical and experimental trials differs from the optimal dose; (v) in addition to the grey literature, three online databases were searched, but it is still possible that some eligible studies were missed; (iv) all included studies had a high risk of bias, which also highlights the need for careful interpretation of the data.

## 6. Conclusions

In this systematic review, we summarized the impact of DDW monotherapy or in combination with chemotherapy on tumour treatment. We showed that DDW suppresses the tumour through regulating the expression of ROS-related genes. The review also indicated the beneficial effects of DDW monotherapy in cancer treatment. It needs to be investigated whether DDW affects the Keap1-Nrf2 signalling pathway. Additionally, more randomized clinical trials are necessary to measure the effects of different concentrations of DDW on cancer treatment in patients. The optimal combination of DDW and various types of chemotherapy should be further studied, which would be a fruitful area for further work. Finally, this study can be considered a guideline for further in-depth clinical research on DDW as an adjuvant therapy in treating patients with cancer.

## Figures and Tables

**Figure 1 nutrients-16-01397-f001:**
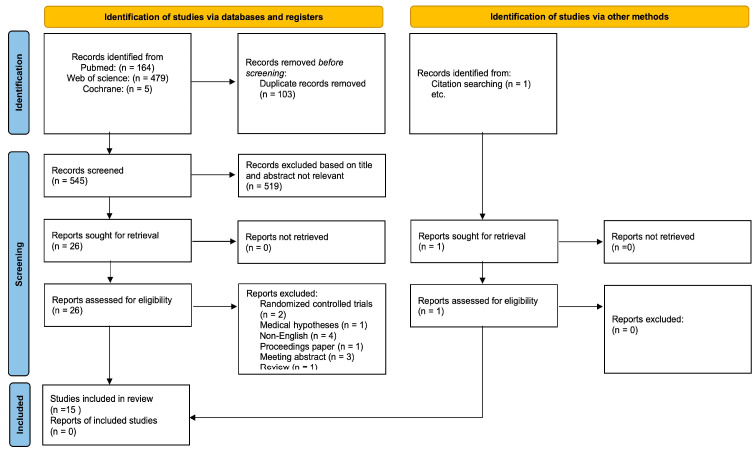
PRISMA 2020 flow diagram for new systematic reviews which included searches of databases, registers and other sources.

**Figure 2 nutrients-16-01397-f002:**
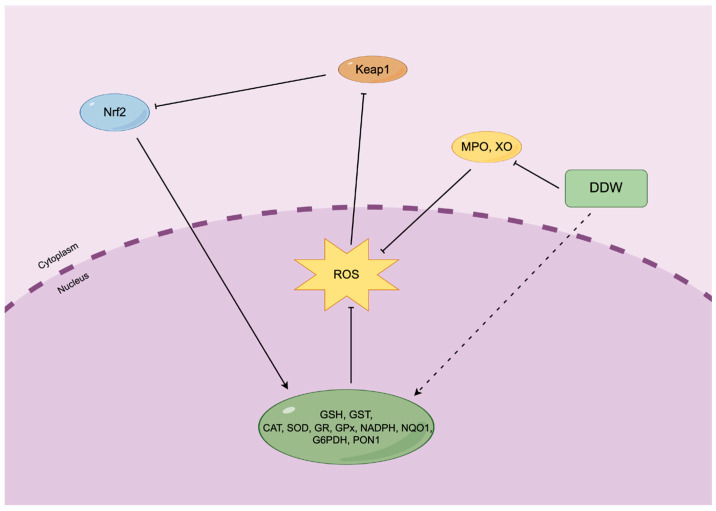
Schematic representation of the Keap1-Nrf2 signalling pathway for gene expression control. (By Figdraw). Excess ROS induces the suppression of Keap1, further disassociating Nrf2 from Keap1 to enter the nucleus. Accumulation of Nrf2 in the cellular nucleus causes antioxidative gene expression, suppressing ROS production. Additionally, DDW controls the expression level of antioxidative genes and inhibits the oxidative genes.

**Figure 3 nutrients-16-01397-f003:**
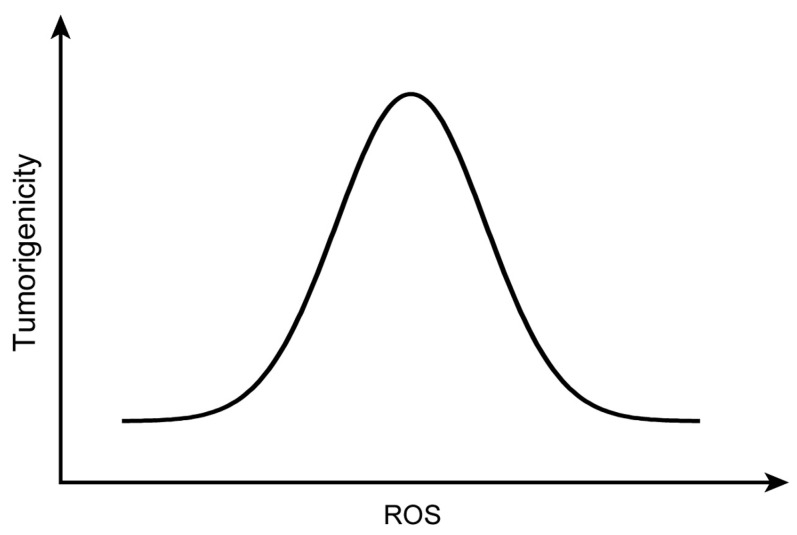
Schematic diagram representing the association between the amount of ROS and tumorigenicity. Initially, the increase in ROS could induce tumorigenesis. When ROS production exceeds the threshold, the formation of the tumour is inhibited. Further increases in ROS will induce programmed cell death in the tumour.

## Data Availability

No new data were created or analysed in this study. Data sharing is not applicable to this article.

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
