# Peer review of "Deuterium-Depleted Water in Cancer Therapy: A Systematic Review of Clinical and Experimental Trials"

_nutrients, 2024, doi:10.3390/nu16091397_

Round 1

Reviewer 1 Report

Comments and Suggestions for Authors

In this manuscript authors systematically reviewed the current literature regarding the potential mechanism of DDW in cancer therapy highlighting that DDW alone or in combination with chemotherapy inhibited cancer progression in several studies. Moreover, authors found that the combination treatment enhances the therapeutic effect on cancer compared with chemotherapeutic monotherapy demonstrating an inhibitory role of DDW. 

Although the manuscript is interesting, it presents some flaws that must be resolved. My comment are listed below.

Introduction: Although the NRF2/KEAP1 signaling plays a pivotal role in this manuscript, it it not even mentioned in the introduction. The multifunctional role of this signaling must be stated since it is involved in the onset and progression of several cancerous and non-cancerous diseases (as also recently reviewed PMID: 37296665, PMID: 36641100)

Figure 1: this figure must contain only the flow diagram without references (that must be added as reference in the references list) and title "PRISMA 2020 flow diagram......"

Table 1 and 2: please modify the format according to the journal style. Moreover, the column "Study design" is useless since the the fourth column is stated the model used in the study. The column "Author" is useless and should be replaced by a column "Reference" showing the number of the reference according to the journal style. 

Table 3 is useless since there is only one study discussed

Subheadings should be written in italic

There are too many subheadings, please reduce them when possible

An accurate revision of typing errors and punctuation is necessary

Abbreviations must be written in full length the first time that are mentioned

Reviewer 2 Report

Comments and Suggestions for Authors

1. I have also used DDW, and it has long been known to be effective against cancer, so it is not new.

2. However, it is very good to see the mechanism not only at the clinical level but also at the research level.

3. Although there are various other substances that suppress cancer, it is necessary to discuss why DDW was chosen this time.

4. This research seems to require ethical review, but what about that?
